# CONDITIONAL TRAJECTORIES IN DIFFUSION MODELS
## MODELING GALAXY EVOLUTION FROM REDSHIFT

## ABSTRACT

In this paper, we present a novel approach for continuous **C**onditional **T**rajectories on Denoising **D**iffusion Probabilistic **M**odels (CTDM). Focusing on physical applications, our model learns to capture the underlying relationship between galaxy images and their redshift values from training data. This enables the simulation of galaxy evolution by conditioning the reverse denoising process on future redshift values. Importantly, this is achieved without requiring multiple images of the same galaxy at different redshifts. We demonstrate that our redshift-conditioned diffusion model learns the marginal distribution of galaxy images at each redshift value. This allows the model to generate realistic galaxy images that reflect the physical changes occurring as galaxies evolve. We derive a smoothness condition for this learned distribution, proving that the model can construct trajectories between galaxy images by incrementally changing redshift during the reverse denoising process. Our approach offers a novel interpretation of the learned diffusion process as a means to simulate galaxy evolution, capturing both visual and physical changes over time. These techniques not only provide deeper insights into the formation and evolution of galaxies but also have broader potential applications in various areas of generative modeling.

## 1 INTRODUCTION

Understanding galaxy formation and evolution is central to astrophysics, yet observational limitations restrict our ability to capture galaxies across cosmic timescales. Redshift-conditioned generative models can help address this challenge by simulating galaxies in underexplored regions, thus thereby offering new insights into galaxy evolution and cosmic structure. Recently, Denoising Diffusion Probabilistic Models (DDPMs) Ho et al. (2020) have emerged as a promising class of generative models, achieving state-of-the-art results in generating high-fidelity images Ho et al. (2020); Nichol & Dhariwal (2021); Dhariwal & Nichol (2021). These DDPMs have been proposed by (Li et al., 2024; Xue et al., 2023; Nguyen et al., 2024; Lastufka et al., 2024) as suitable models for modeling galaxy evolution. However conditioning these models on continuous attributes such as redshift proves to be difficult and is the main focus of this work.

Our work builds on the concept of conditional generation by focusing on continuous attributes and exploring how to construct smooth transitions in the latent space as the conditioning variable $z$ changes. Although previous research has demonstrated that diffusion models can generate high-quality images, relatively few studies have explored the behavior of these models when the conditioning variable is continuous. In this regard, our work shares similarities with efforts to enforce smoothness in latent spaces Kingma & Welling (2014), particularly in the context of variational autoencoders (VAEs) and generative adversarial networks (GANs).

Conditional trajectory-based generation has been explored in related areas such as VAEs, where latent space interpolation is commonly used to demonstrate the continuity of the learned space Esser et al. (2021). However, due to their stochastic nature, diffusion models offer a distinct framework for generating such trajectories. Additionally, our empirical validation of the smoothness assumption relates to studies on the stability of generative models under small perturbations Arjovsky et al. (2017), which highlight the importance of enforcing stability in high-dimensional generative tasks. We specifically focus on continuous changes in the conditioning variable and evaluate their impact on the galaxy morphology to determine whether the evolution proposed by our model is physically plausible (Sec. 5).

It is not possible to capture the same galaxy at multiple redshifts because we cannot go backward or forwards in time, resulting in the absence of ground truth for comparing the model's results. To address this, we test the model's stability in producing trajectories and verify whether the redshift of the generated image trajectories corresponds to the redshifts on which the model was conditioned. Additionally, we empirically validate when the model satisfies the smoothness of conditioning assumptions and when it does not (Sec. 6.2).

## 2    RELATED WORK

Recent efforts by Li et al. (2024); Smith et al. (2022) have applied diffusion models in astronomy by discretizing continuous redshift values to fit the discrete-time framework of these models. However, this discretization inherently leads to information loss, limiting the model's ability to accurately learn the continuous distribution $p(X^z \mid z)$ and affecting the precision of generated galaxy images conditioned on redshift. Similar approaches, such as those by Xue et al. (2023), have explored the use of DDPMs for Point Spread Function (PSF) deconvolution, but their methods do not address the limitations of discrete stepwise conditioning. Lanusse et al. (2021) and Margalef-Bentabol et al. (2020) employed Generative Adversarial Networks (GANs) with redshift as a conditional input to generate synthetic galaxy images, simulating visual characteristics across different distances and observational scenarios. However, these GANs struggle with mode collapse, and their benchmarks rely on perceptual scores rather than galaxy morphology, which is tied to the physics of galaxy evolution. Recent two stage approaches using normalizing flows have been proposed by Nguyen et al. (2024) on a small discrete physical parameter space to inject physics information into a DDPM. However, this approach comes at the cost of using computational normalizing flows and can only hand a small number of parameters. Lastufka et al. (2024) performed a recent analysis on the utility of recent vision foundation models to capture the distribution of galaxies but has noted that it is difficult to integrate standard vision models with the low-resolution modalities of galaxy-based data and that exceptional care must be taken if one wants to adapt them to such tasks.

## 3    CONTRIBUTIONS

To overcome these limitations, we propose a novel adaptation of DDPMs specifically tailored for generating galaxy images across a continuous range of redshifts without the need for discretization or introducing a secondary redshift encoding model. Our main contributions are as follows:

- We develop a new approach that directly conditions the DDPM on continuous redshift values, significantly enhancing the model's accuracy and fidelity.

- We demonstrate that under certain smoothness and bounded gradient conditions, the model can construct image trajectories even without observing the same image conditioned at multiple values of $z$.

- Our findings show that the model can implicitly learn the morphological characteristics of galaxies without explicit input regarding these attributes, suggesting that redshift alone is predictive of galaxy morphology.

- To our knowledge, this is the first work demonstrating a potential approach to dynamically understand galaxy evolution through redshift and image alone.

### 3.1    DATA

For our analysis, we employ a subset of the *Hyper Suprime-Cam Galaxy Dataset* curated by Do et al. Do et al. (2024), which is publicly accessible at Zenodo (GalaxiesML: `https://zenodo.org/records/11117528` CC-BY 4.0). This dataset is based on the data released by the Hyper Suprime-Cam survey, as detailed by Aihara et al. Hiroaki Aihara & et al. (2019). It comprises 286,401 galaxies, spanning redshifts from 0 to 4. The redshift is related to the distance of the galaxy and the time the light was emitted. For example, light from a galaxy at a redshift of $z = 1$ was emitted about 7.8 billion years ago. Each galaxy is represented by images taken in five visible wavelength bands—$(g, r, i, z, y)$ filters. We use the $64 \times 64$ pixel images from GalaxiesML. The dataset includes accurate spectroscopic measurements of each galaxy's true redshift (or distance

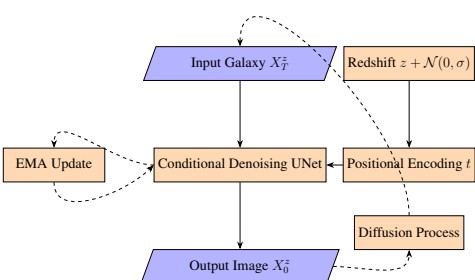

Figure 1: Model Architecture

from Earth). Due to the selection process, the dataset exhibits a bias toward lower redshifts, with approximately 92.8% of the galaxies having redshifts less than 1.5 (∼9.5 billion years ago). We adhere to the training and testing split proposed by Li et al. Li et al. (2024), resulting in a training set comprising 204,513 images and a testing set containing 40,914 images.

## 4 METHODS

In this section, we highlight two key insights that form the basis of our approach. First, since galaxies are characterized by continuous attributes such as redshift, it is beneficial to use a DDPM with continuous conditioning. This approach allows the model to more effectively capture the smooth variations in galaxy properties over different redshift values than discrete conditioning methods.

Second, we demonstrate that by utilizing only the image data and the continuous attribute (redshift), a DDPM can learn key galaxy morphological characteristics without being explicitly provided with that information. This indicates that the model inherently learns underlying physical properties of galaxies as a function of redshift, enabling it to generate realistic galaxy images that reflect morphological changes over cosmic time.

In Section 5 we will use these insights to demonstrate how to evolve a galaxy via redshift conditional trajectory reconstruction methods.

### 4.1 LEARNING THE CONTINUOUS CONDITIONING OF DDPMs

Utilizing DDPMs Ho et al. (2020), we introduce a novel approach to learn the conditional distribution $p(X^z \mid z)$ by integrating redshift values into the U-Net architecture's time steps Li et al. (2024); Smith et al. (2022). To prevent model overfitting and ensure learning is concentrated within a Gaussian neighborhood around specific redshifts $z$, Gaussian noise $\mathcal{N}(0, \sigma^2)$ is added to the redshifts during training, enhancing the model's ability to interpolate between nearby redshifts. Our Conditional Denoising U-Net starts with a noisy initial galaxy image $X_T^z$ and, through iterative denoising informed by both time step and the adjusted redshifts, aims to produce a clean galaxy image $X_0^z$. To additionally stabilize the training, we implement an Exponential Moving Average (EMA) Karras et al. (2024) and adhere to a standard variance schedule Ho et al. (2020); Song et al. (2020) to balance noise addition and preserve data structure.

The model's diffusion process starts with $64 \times 64$ pixel galaxies images with 5 channels, which are passed to a noising schedule across $1,000$ time steps, linearly interpolating noise levels from a Beta Start of $1 \times 10^{-4}$ to a Beta End of 0.02. Training utilizes Huber Loss for its robustness to outliers, gradient clipping with a maximum norm of 1.0, and an AdamW optimizer Loshchilov & Hutter (2020) set to a learning rate of $2 \times 10^{-5}$. Redshifts are perturbed with Gaussian noise (std dev 0.01) to prevent overfitting and improve generalization. Our U-Net model, equipped with self-attention layers, varies channels by resolution stage and includes 4 attention heads with layer normalization and GELU activation Hendrycks & Gimpel (2016), applied before and after attention. Temporal and conditional redshift information is encoded using sinusoidal positional encoding of the time step $t$, transformed into a 256-dimensional vector. This vector is further modified by adding

Gaussian noise to the redshift value $z + \mathcal{N}(0, 0.01)$, prior to being fed into the U-Net (refer to 4.1). The model was trained on a single NVIDIA A6000 GPU. *Anonymous Code Link:* `https://anonymous.4open.science/r/Generative-Modeling-6BFC/README.md`

## 5 EVALUATING THE DDPMs ABILITY TO IMPLICITLY CAPTURE GALAXY MORPHOLOGY

Our evaluation focuses on the measured physical attributes of galaxies to gauge the physical consistency of our generated images, which involve five color filters $(g, r, i, z, y)$. While perceptual quality metrics like Fréchet Inception Distance (FID) Heusel et al. (2017) and Inception Score (IS) Salimans et al. (2016) indicate general similarity to true images, they fail to assess critical morphological properties of galaxies and their evolution over time. Our evaluation involves generating synthetic images conditioned on redshifts from the test dataset and comparing to physical properties that astronomers typically use to characterize galaxies (e.g. Conselice, 2014a), such as the shape (ellipticity, semi-major axis), size (isophotal area), and brightness distribution (Sersic index). Furthermore, using the CNNRedshift predictor established by Li et al. Li et al. (2024), we assess the redshift accuracy against the ground truth, utilizing the redshift loss from Nishizawa et al. (2020). This redshift predictor was trained on real galaxy images using spectroscopic ground truth and produces good predictions on real data (Fig. 2). These comparisons help verify the physical plausibility of the diffusion model's output.

### 5.1 REDSHIFT PREDICTION

We find that the generated images have redshift predictions that are in good agreement with the redshift that they were generated with as evaluated by the CNNRedshift predictor (Li et al., 2024) (Fig. 2). The DDPM produces images with redshift predictions that have slightly larger scatter than with real images, but follows the 1:1 line between conditioned redshift and predicted redshift well up to a redshift about 2. Redshifts beyond 2 are challenging because these redshifts represent less than 2% of the training dataset.

### 5.2 MODELING THE PHYSICAL CHARACTERISTICS OF GALAXIES

We calculate standard metrics for both the test data and the DDPM-generated images, which are conditioned on the test data's redshifts. Our findings confirm that the DDPM successfully learns the physical characteristics of galaxies-such as the ellipticity, semi-major axis, Sersic index, and isophotal area even though these attributes were never explicitly provided to the model. When comparing the frequencies of each metric between the DDPM and the true distribution, we see in Fig. 3 that the overall shape of the distributions is very close. Thus for any conditioned redshift of the model, the image produced is a physically plausible galaxy.

Moreso, Fig. 4 illustrates that for each redshift bin, the mean values (represented by red dots) of each metric for DDPM-generated galaxies closely match the means of the true test distribution (blue dots). The ranges of these metrics generally fall within the true distribution's ranges. This suggests that the DDPM model is able to associate redshifts with morphological characteristics of galaxies observed at that redshift. For example, the galaxies tend to be more compact at higher redshifts but the distribution of ellipticity does not change much with redshift, consient with the testing dataset.

Recall that Fig. 2 indicates a greater variance in detected redshifts, but Fig.3 and 4 suggest the galaxies structures are physically accurate. What this entails is that even though we expect the generated images of galaxies to have appropriate morphology, we should anticipate the model to produce a broader range of generated images for higher redshift values, potentially blending characteristics from neighboring redshift value. This in turn contributes to more variance in redshift. This effect is evident in Fig. 5, where the model generates images that display increased diversity and variability. This high variance in the predicted conditioned output of the model is also a good indicator of when to expect the model to fail at conditional trajectory construction as seen in Fig. 7. Sec. 6 provides a further detailed analysis of trajectory construction and C.2 has derived results and conditions for successful trajectory reconstruction.

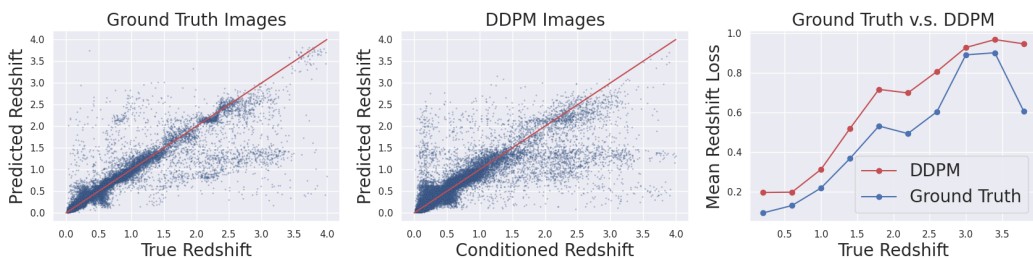

Figure 2: From left to right, the figure displays: 1) a scatter plot comparing predicted redshifts to true redshifts for ground truth images, 2) a similar scatter plot for DDPM-generated images, 3) a plot of true redshift versus mean redshift loss, highlighting the performance accuracy across the redshift range.

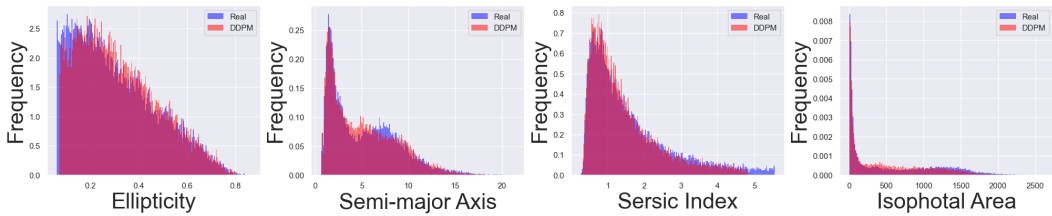

Figure 3: From left to right, the figure displays histograms comparing the frequency distribution of DDPM-generated and real galaxies in terms of 1) ellipticity, 2) semi-major axis, 3) Sersic index, and 4 isophotal area).

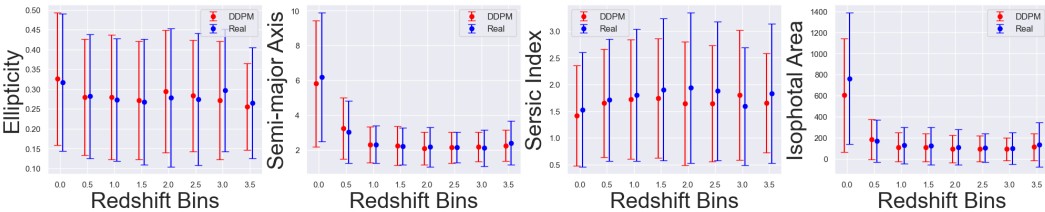

Figure 4: From left to right, the figure displays $95\%$ CIs comparing DDPM-generated and real galaxies across redshift bins: 1) ellipticity, 2) semi-major axis, 3) Sersic index, and 4 isophotal area).

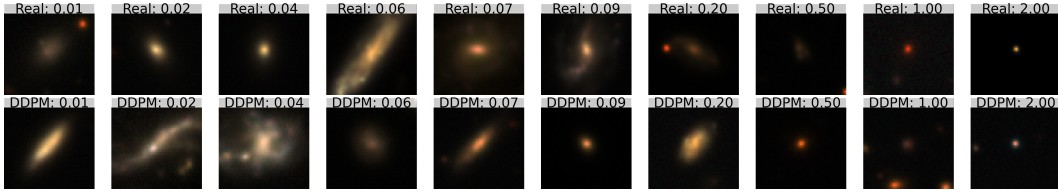

Figure 5: (Top) Real galaxies and corresponding redshifts and (Bottom) DDPM generated galaxies. Both rows correspond to respective redshifts.

## 6 CONSTRUCTING CONDITIONAL TRAJECTORIES

Once we have determined that our continuously conditioned DDPM accurately captures the data distribution, we focus our efforts on evolving galaxies from the test distribution through their redshift by constructing image trajectories (see Algorithm 1 for exact details on trajectory reconstruction).

Note that for astronomical data, it is not possible to observe the same galaxy at multiple redshift values. Therefore, our dataset comprises many different galaxies—possibly sharing similar physical characteristics—at different redshifts. To understand the evolution of a galaxy, we propose that if the continuously conditioned DDPM has learned the distribution $p(X^z|z)$ sufficiently well, and under suitable assumptions (see A.1.1) that we expect to hold for galaxy data, then we can construct a galaxy evolving through redshift. Specifically, we learn to reconstruct a smooth trajectory in $z$: $X^z, X^{z+\Delta z}, X^{z+2\Delta z}, \ldots$. To our knowledge, this is the first attempt to achieve this using galaxy images alone. The formal methods and algorithms are derived in A.1.1 and A.1.2, but intuitively the process works as follows:

Let $X^z$ denote an image conditioned on the redshift $z$. Assume that a diffusion model has been trained to recover the marginal distribution $p(X^z)$ for each $z$, using a conditional denoising process based on a continuous variable $z$. For any image $X^z$, we can construct the next step in a galaxy's evolution, $X^{z+\Delta z}$, by:

1. Adding Gaussian noise to $X^z$ according to the forward diffusion process.
2. Applying the reverse diffusion process conditioned on $z + \Delta z$.

Again, the formal algorithm and derivation are fully described in A.1.1.

### 6.1 ASSUMPTIONS

To enable the reconstruction of smooth trajectories $X^z, X^{z+\Delta z}, X^{z+2\Delta z}, \ldots$ using a DDPM, we rely on several key assumptions. These assumptions are motivated by the physical characteristics of galaxies and the nature of astronomical data.

First, we assume that the model has learned the diffusion process correctly. Given the forward process $q(X_t^z|X_{t-1}^z)$, the model learns the reverse process $p_\theta(X_{t-1}^z|X_t^z, z)$, where $t$ represents discrete time steps in the diffusion process, parameterized by noise levels $\beta_t$, and the reverse process is conditioned on the redshift $z$. This assumption ensures that the DDPM can effectively model the data distribution at each redshift level and is confirmed in Sec. 5.

Second, we assume smoothness in the conditional distribution $p(X|z)$ with respect to the redshift $z$. Specifically, for any small $\Delta z$, the Kullback-Leibler (KL) divergence between $p(X|z)$ and $p(X|z + \Delta z)$ is small, tending to zero as $\Delta z \to 0$:

$$\text{KL}(p(X|z)\|p(X|z+\Delta z)) \to 0 \quad \text{as} \quad \Delta z \to 0.$$

This smoothness assumption reflects the gradual changes in galaxy images as a function of redshift, implying that galaxies at nearby redshifts have similar visual and spectral properties (see Sec. 5, Figs. 2, 3, 4) Conselice (2014b).

Third, we assume that the gradient of the learned reverse process with respect to $z$ is bounded in $z$-space. That is, there exists a constant $C > 0$ such that for all $t$,

$$\|\nabla_z \mu_\theta(X_t^z, t, z)\| \leq C,$$

where $\mu_\theta$ represents the estimated mean in the reverse diffusion process. This bounded gradient ensures stability in the model's predictions as we vary $z$, preventing abrupt changes that could disrupt the smoothness of the trajectory. See A.1.1 and A.1.2 for exact formulations of these assumptions and the proof of their necessity.

These assumptions are intuitive when considering the structure of galaxies. Galaxies evolve slowly over cosmological timescales, and their observable properties change gradually with redshift due to factors like cosmic expansion and redshift of light Peebles (1993). Therefore, small changes in redshift correspond to subtle changes in galaxy images, supporting the smoothness and boundedness assumptions. Under these conditions, the DDPM is expected to reconstruct smooth trajectories of galaxy evolution through redshift. A formal proof of this claim is provided in Appendix A.1.2. The following section empirically verifies these assumptions.

## 6.2 Evaluating Conditional Trajectories

From the test set, we sample real images and evolve them in the positive direction of redshift using Algorithm 1. The image trajectories in Fig. 8 can be subtle; though we expect to see a subtle change of the red hue as we increase the redshift, we typically do not observe dramatic visual shape changes but expect changes in the spectral intensities. To verify that the model's results are as expected, we use the CNNRedshift regressor Li et al. (2024) to predict the perturbed redshifts and determine if they are aligned with the redshifts we conditioned the trajectory on.

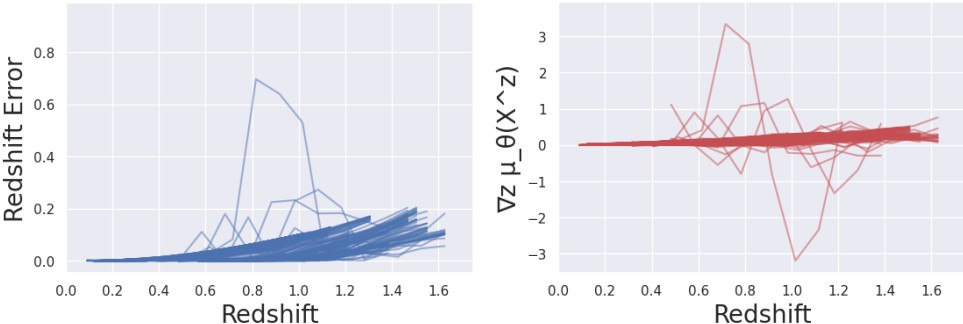

Figure 6: (Left) Images from the testset with redshifts ranging from 0.01 to 0.1 are evolved via Algorithm 1. Redshift predictions are then taken for 10 generated images at steps of size $\Delta z = 0.2$ and the error between the conditioned redshift and the predicted redshift are plotted. (Right) The $z$-gradients of each image of the trajectory evaluated under the denoising model are computed and we see that the gradients remain fairly constant near zero.

As seen in Fig. 6 (Left), when constructing a conditional trajectory: $X^z, X^{z+\Delta z}, X^{z+2\Delta z}, \ldots$ see that the difference between the conditioned redshift $z + n\Delta z$ and the predicted redshift $\hat{z}$ is relatively small and the error grows gradually (with the exception of a few outliers). This suggests that smoothness in $z$ holds and that we expect that generated sequence to be reasonably accurate. Additionally Fig. 6 (Right) shows a plot of the gradients of $\mu_\theta(X^z)$ with respect to $z$. We see that the trajectory gradients remain stable and close to 0 suggesting that the model is satisfying our hypothesized assumptions 6.1 A.1.1. In otherwords for the redshift range of $z \in (0, 1.6)$ there were sufficiently many pairs $(X^z, z)$ to be able to construct trajectories for any $X^z$ despite not have information of $X^z$ at different redshift values.

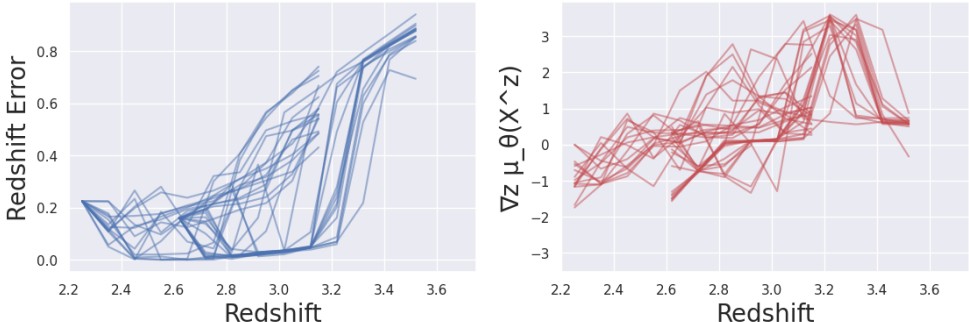

Figure 7: (Left) Images from the testset with redshifts ranging from 2.2 to 2.6 are evolved via Algorithm 1. Redshift predictions are then taken for 10 generated images at steps of size $\Delta z = 0.2$ and the error between the conditioned redshift and the predicted redshift are plotted. (Right) The $z$-gradients of each image of the trajectory evaluated under the denoising model are computed and we see that the gradients are not constant.

Recall that $92.8\%$ of the training data (see Sec. 3.1) has a redshift value of less than $1.5$. As we saw in Fig. 2, the DDPM model tends to perform poorly on correctly associating appropriate images to

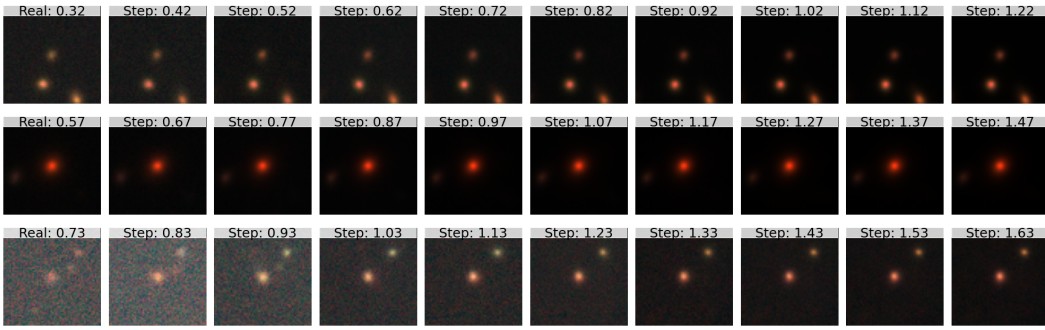

Figure 8: Real images and their corresponding trajectories from Algorithm 1. Additional image trajectories can be found in C.2 Fig. 11. The left-most image depicts the real galaxy image and real redshift from the test set. Each step to the right indicates the image produces from by conditioning on the redshift $+.10$. Note that visually the changes in redshift are sublte, but detectable as mentioned in Sec. 6.2 and as indicated in Fig. 6.

large redshifts. This is also reflected in Fig. 7. Since the region of redshift $z \in (2.2, 2.6)$ has a sparse number of examples, we see that the the errors in redshift (Left) fail to have a gradual progression suggesting the trajectories fail the smoothness in $z$ assumption and that the gradients (Right) are not constant and appear to be increase as the we increment the trajectory.

## 7   LIMITATIONS

Despite the promising results of our model, several limitations need to be acknowledged. One of the key challenges is that galaxies do not evolve in isolation. Our model currently treats each galaxy independently, failing to account for the complex interactions between galaxies and their environments, such as mergers or gravitational interactions. These interactions play a significant role in galaxy evolution, and ignoring them may limit the physical accuracy of the generated trajectories.

Additionally, while our model successfully generates realistic galaxy images conditioned on redshift, the denoising process might inadvertently remove noise that encodes important physical information in later stages of galaxy evolution, particularly in video sequences. This smoothing effect could reduce the overall realism of the generated data, particularly when simulating high-redshift galaxies. As a result, there is a risk that the model may introduce artifacts as the denoising process progresses, potentially compromising the fidelity of galaxy structures at later stages. Additionally metrics such as Sersic index, ellipticity, and isophotal area can potentially have higher variance when constructing these conditional trajectories, since the conditioning is solely based on redshift alone (see Fig. 9).

Moreover, the model's performance is notably less reliable at higher redshifts, where the training data is sparse. This limitation indicates that the model struggles to capture the full diversity of galaxy morphologies at these redshifts, leading to increased variability in the generated images and less accurate redshift conditioning.

## 8   CONCLUSION

In this paper, we introduced a novel approach for constructing continuous conditional trajectories via Denoising Diffusion Probabilistic Models (DDPMs) to simulate the evolution of galaxies through redshift. By conditioning the reverse denoising process on continuous redshift values, our model effectively learns the marginal distribution $p(X^z \mid z)$ of galaxy images at each redshift, enabling the generation of realistic images that reflect the physical changes occurring as galaxies evolve.

Our method leverages a smoothness condition in the learned distribution, allowing for the construction of image trajectories by incrementally changing the redshift during the reverse diffusion process. Importantly, this is achieved without requiring multiple images of the same galaxy at different redshifts, a common limitation in astronomical datasets.

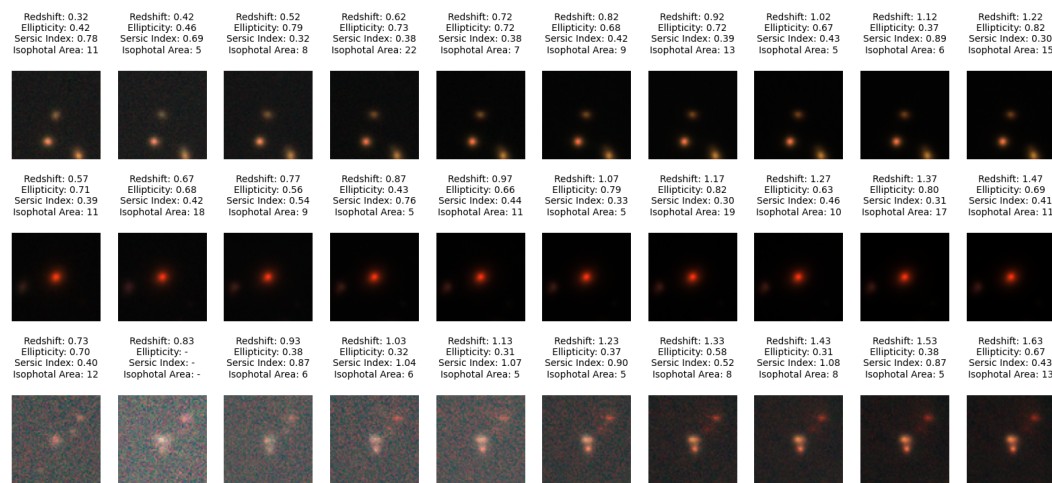

Figure 9: Real images and their corresponding trajectories from Algorithm 1. Viewing physcial metrics as the we construct a a trajectory based on the real image (Left). The model is only conditioned on redshift, while metrics such as Sersic Index, Ellipticity, Isophotal area may be assocated with certain redshift bins as in Fig. 3, the variance of these metrics can be higher for these trajectories which are out of distribution. Note that depending on noise-to-signal, some metrics computations are not computed and are left blank.

Through extensive evaluations, we demonstrated that our continuously conditioned DDPM captures key morphological characteristics of galaxies as a function of redshift, even though these attributes were not explicitly provided during training. The generated images not only exhibit plausible physical properties but also maintain consistency in morphological metrics such as ellipticity, semi-major axis, Sersic index, and isophotal area when compared to real galaxy images.

We also empirically verified the model's ability to construct smooth trajectories in redshift space, validating our theoretical assumptions about smoothness and bounded gradients. Our results show that the model performs well within redshift ranges that are well-represented in the training data. However, we observed limitations at higher redshifts due to data sparsity, indicating areas for future improvement.

A significant challenge in modeling galaxy evolution is the lack of ground truth data for observing the same galaxy at multiple redshift values. Since we cannot track individual galaxies over cosmic timescales, we must rely on our assumptions and empirical validations to ensure the plausibility of the generated evolutionary trajectories. Future work should focus on comparing our model's trajectories with physics-based simulations, such as hydrodynamical or semi-analytic models, to further validate the physical realism of the generated images. Integrating these simulations could provide a benchmark for assessing the accuracy of our approach and help refine the model to better capture the complexities of galaxy evolution.

Our approach offers a new avenue for simulating galaxy evolution, providing a dynamic representation that can enhance our understanding of cosmic structures over time. Beyond astrophysics, the techniques developed in this work have potential applications in other domains where modeling continuous transformations conditioned on scalar variables is valuable, such as computer vision and graphics.

Future work could focus on extending the model to incorporate additional physical parameters, improving performance at higher redshifts by augmenting the training dataset, and exploring the integration of this method with observational data to aid in astronomical discoveries.

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

## A APPENDIX

### A.1 CONTINUOUS TRAJECTORY RECONSTRUCTION

We provide the following assumptions and conditions that allow for conditional trajectory reconstructions.

#### A.1.1 ASSUMPTIONS

Under the following three assumptions, a DDPM can learn to reconstruct a smooth trajectory: $X^z, X^{z+\Delta z}, X^{z+2\Delta z}, \ldots$.

1. **Learned Diffusion Process**: Given the forward process $q(X_t^z|X_{t-1}^z)$, the model learns the reverse process $p_\theta(X_{t-1}^z|X_t^z, z)$, where $t$ represents discrete time steps in the diffusion process. These are parameterized by noise levels $\beta_t$, and the reverse process is conditioned on $z$.

2. **Smoothness Assumption**: The conditional distribution $p(X|z)$ is smooth in $z$. Specifically, for any small $\Delta z$, the Kullback-Leibler (KL) divergence between $p(X|z)$ and $p(X|z + \Delta z)$ is small, i.e.,

$$\text{KL}(p(X|z)\|p(X|z + \Delta z)) \to 0 \quad \text{as} \quad \Delta z \to 0.$$

3. **Bounded Gradient in $z$-space**: The gradient of the learned reverse process with respect to $z$, $\nabla_z \mu_\theta(X_t^z, t, z)$, is bounded for all $t$, i.e., there exists $C > 0$ such that:

$$\|\nabla_z \mu_\theta(X_t^z, t, z)\| \leq C.$$

### A.1.2 Proof of Continuous Trajectory Reconstruction

Conditions (A.1.1.1) is immediate, since we require the learned diffusion model to be able to accurately denoise any image in it's distribution. Under the smoothness (A.1.1.2) assumption and bounded gradients (A.1.1.3), we analyze the difference between the reverse processes conditioned on $z$ and $z + \Delta z$.

At each reverse diffusion step, the mean of the reverse process conditioned on $z$ is given by:

$$\mu_\theta(X_t^z, t, z) = \frac{1}{\sqrt{\alpha_t}} \left( X_t^z - \frac{\beta_t}{\sqrt{1 - \bar{\alpha}_t}} \epsilon_\theta(X_t^z, t, z) \right)$$

Similarly, the mean conditioned on $z + \Delta z$ is:

$$\mu_\theta(X_t^z, t, z + \Delta z) = \frac{1}{\sqrt{\alpha_t}} \left( X_t^z - \frac{\beta_t}{\sqrt{1 - \bar{\alpha}_t}} \epsilon_\theta(X_t^z, t, z + \Delta z) \right)$$

The difference in the means due to the change in $z$ is:

$$\Delta \mu_t = \mu_\theta(X_t^z, t, z + \Delta z) - \mu_\theta(X_t^z, t, z)$$
$$= \frac{1}{\sqrt{\alpha_t}} \left( -\frac{\beta_t}{\sqrt{1 - \bar{\alpha}_t}} \left( \epsilon_\theta(X_t^z, t, z + \Delta z) - \epsilon_\theta(X_t^z, t, z) \right) \right)$$

Assuming that $\epsilon_\theta(X_t^z, t, z)$ is smooth (A.1.1.2) with respect to $z$, we can perform a first-order Taylor expansion around $z$:

$$\epsilon_\theta(X_t^z, t, z + \Delta z) \approx \epsilon_\theta(X_t^z, t, z) + \nabla_z \epsilon_\theta(X_t^z, t, z) \cdot \Delta z$$

Substituting back into $\Delta \mu_t$, we get:

$$\Delta \mu_t \approx \frac{1}{\sqrt{\alpha_t}} \left( -\frac{\beta_t}{\sqrt{1 - \bar{\alpha}_t}} \left( \nabla_z \epsilon_\theta(X_t^z, t, z) \cdot \Delta z \right) \right)$$
$$= -\frac{\beta_t}{\sqrt{\alpha_t(1 - \bar{\alpha}_t)}} \left( \nabla_z \epsilon_\theta(X_t^z, t, z) \cdot \Delta z \right)$$

Therefore, the difference in the reverse process mean is proportional to $\nabla_z \epsilon_\theta(X_t^z, t, z) \cdot \Delta z$.

Since the next state in the reverse process is sampled as:

$$X_{t-1}^{z+\Delta z} = \mu_\theta(X_t^z, t, z + \Delta z) + \tilde{\beta}_t \mathbf{z}, \quad \text{with} \quad \mathbf{z} \sim \mathcal{N}(0, \mathbf{I}),$$

and similarly for $X_{t-1}^z$, the difference in the next states is:

$$\Delta X_{t-1} = X_{t-1}^{z+\Delta z} - X_{t-1}^z$$
$$= \mu_\theta(X_t^z, t, z + \Delta z) - \mu_\theta(X_t^z, t, z)$$
$$= \Delta \mu_t$$

Therefore,

$$\Delta X_{t-1} \approx -\frac{\beta_t}{\sqrt{\alpha_t(1 - \bar{\alpha}_t)}} \left( \nabla_z \epsilon_\theta(X_t^z, t, z) \cdot \Delta z \right)$$

Taking the norm of $\Delta X_{t-1}$, and using the bounded gradient assumption (6.1.3) (there exists $C > 0$ such that $\|\nabla_z \epsilon_\theta(X_t^z, t, z)\| \leq C$), we have:

$$\|\Delta X_{t-1}\| \leq \frac{\beta_t}{\sqrt{\alpha_t(1 - \bar{\alpha}_t)}} C \|\Delta z\|$$

Since $\beta_t$, $\alpha_t$, and $\bar{\alpha}_t$ are known scalar quantities from the noise schedule, we can denote:

$$C_t = \frac{\beta_t}{\sqrt{\alpha_t(1 - \bar{\alpha}_t)}}\, C,$$

so the bound becomes:

$$\|\Delta X_{t-1}\| \leq C_t \|\Delta z\|$$

By iterating this bound over all time steps $t$, the cumulative error remains controlled. Therefore, small changes in $z$ lead to small changes in the trajectory, validating the method. Consequently the produced sequence of galaxies should correspond to their perturbed redshift values and shouldn't dramatically fluctuate from their predicted redshift (See Fig. 6). The algorithm for this process is described as follows:

---

**Algorithm 1:** Trajectory Construction in Continuous Conditional Diffusion Models

---

**Input:** Initial image $X^z$, initial condition $z$, step size $\Delta z$, number of steps $N$, trained diffusion
      model $p_\theta$

**Output:** Sequence of images $\{X^{z+n\Delta z}\}_{n=1}^N$

**for** $n \leftarrow 1$ **to** $N$ **do**

    **Forward Diffusion (Adding Noise):**

    Obtain noisy image $X_T^{z+(n-1)\Delta z}$ by adding Gaussian noise to $X^{z+(n-1)\Delta z}$:

    **for** $t \leftarrow 1$ **to** $T$ **do**

        Sample

        $X_t^{z+(n-1)\Delta z} \sim q(X_t|X_{t-1}^{z+(n-1)\Delta z}) = \mathcal{N}\left(X_t^{z+(n-1)\Delta z}; \sqrt{1 - \beta_t}X_{t-1}^{z+(n-1)\Delta z}, \beta_t\mathbf{I}\right)$

    **end**

    **Reverse Diffusion (Denoising):**

    Initialize $X_T^{z+n\Delta z} \leftarrow X_T^{z+(n-1)\Delta z}$

    **for** $t \leftarrow T$ **to** $1$ **do**

        Compute mean $\mu_\theta(X_t^{z+n\Delta z}, t, z + n\Delta z)$:

        $\mu_\theta = \frac{1}{\sqrt{\alpha_t}}\left(X_t^{z+n\Delta z} - \frac{\beta_t}{\sqrt{1 - \bar{\alpha}_t}}\epsilon_\theta(X_t^{z+n\Delta z}, t, z + n\Delta z)\right)$

        Sample $X_{t-1}^{z+n\Delta z} \sim \mathcal{N}\left(X_{t-1}^{z+n\Delta z}; \mu_\theta, \tilde{\beta}_t\mathbf{I}\right)$

    **end**

    Set $X^{z+n\Delta z} \leftarrow X_0^{z+n\Delta z}$

**end**

**return** $\{X^{z+n\Delta z}\}_{n=1}^N$

---

# B ARCHITECTURE AND TRAINING DETAILS

## B.1 UNET ARCHITECTURE

The UNet model is employed as the backbone for the denoising process in the DDPM. The model is conditioned on the time step $t$ and the redshift $z$. The detailed layer configuration for the UNet is provided in Table 1.

## B.2 DIFFUSION PROCESS

The diffusion process is defined by a noising schedule that gradually adds noise to the input images over a fixed number of time steps. The model is trained to reverse this process and denoise the images. The parameters for the diffusion process are as follows:

- **Noise Steps**: 1000
- **Beta Start**: $1 \times 10^{-4}$
- **Beta End**: 0.02
- **Image Size**: 64 x 64 pixels (5 channels)

| Layer | Input Channels | Output Channels | Other Parameters |
|---|---|---|---|
| DoubleConv (Initial) | 5 | 64 | Kernel: 3x3, Padding: 1, Activation: GELU, GroupNorm |
| Down1 | 64 | 128 | Embedding Dim: 256, MaxPool: 2x2, Residual: True |
| Down2 | 128 | 256 | Embedding Dim: 256, MaxPool: 2x2, Residual: True |
| Down3 | 256 | 256 | Embedding Dim: 256, MaxPool: 2x2, Residual: True |
| Bottleneck 1 | 256 | 512 | Kernel: 3x3, Padding: 1, Activation: GELU, GroupNorm |
| Bottleneck 2 | 512 | 512 | Kernel: 3x3, Padding: 1, Activation: GELU, GroupNorm |
| Bottleneck 3 | 512 | 256 | Kernel: 3x3, Padding: 1, Activation: GELU, GroupNorm |
| Up1 | 512 | 128 | Embedding Dim: 256, Upsample: 2x2, Residual: True |
| Up2 | 256 | 64 | Embedding Dim: 256, Upsample: 2x2, Residual: True |
| Up3 | 128 | 64 | Embedding Dim: 256, Upsample: 2x2, Residual: True |
| Output Conv | 64 | 5 | Kernel: 1x1 |

Table 1: UNet Layer Configuration

The noise schedule is calculated using a linear interpolation between 'Beta Start' and 'Beta End', across the defined number of noise steps.

### B.3 EXPONENTIAL MOVING AVERAGE (EMA)

To stabilize the training, an Exponential Moving Average (EMA) of the model parameters is maintained. The EMA helps in smoothing out the updates to the model parameters and is especially useful in the later stages of training.

- **EMA Beta**: 0.995
- **EMA Start Step**: 2000

The EMA parameters are updated as:

$$\text{EMA Weight} = \beta \times \text{Old Weight} + (1 - \beta) \times \text{New Weight}$$

### B.4 TRAINING CONFIGURATION

The model is trained using the Huber Loss (Smooth L1 Loss), which is robust to outliers and provides a balance between L1 and L2 loss. The key training parameters are:

- **Loss Function**: Huber Loss (Smooth L1 Loss) with $\delta = 1.0$
- **Gradient Clipping**: Maximum Norm = 1.0
- **Optimizer**: AdamW with appropriate learning rate of $2 \times 10^{-5}$

During training, the labels are perturbed by adding Gaussian noise with a standard deviation of 0.01, and they are clamped to ensure they remain within the valid range [0, 4]. This helps prevent overfitting and allows the model to generalize better to unseen data.

### B.5 SELF-ATTENTION MECHANISM

The UNet model incorporates self-attention layers to better capture long-range dependencies within the images. The self-attention mechanism operates over the feature maps at different resolutions and is defined with the following parameters:

- **Channels**: Varies depending on the resolution stage (64, 128, 256, etc.)
- **Attention Heads**: 4
- **Layer Normalization**: Applied before and after attention with GELU activation.

### B.6 POSITIONAL ENCODING

The temporal information is embedded into the model using positional encoding. The encoding uses sinusoidal functions to encode the time step $t$ into a fixed-dimensional vector.

- **Time Dimension**: 256
- **Encoding Function**: Sinusoidal encoding with alternating sine and cosine functions.

The noised redshift $z + \mathcal{N}(0, 0.01)$ is then added to the encoded time step before being passed to the UNet.

# C IMAGE GENERATIONS

## C.1 GENERATED GALAXIES

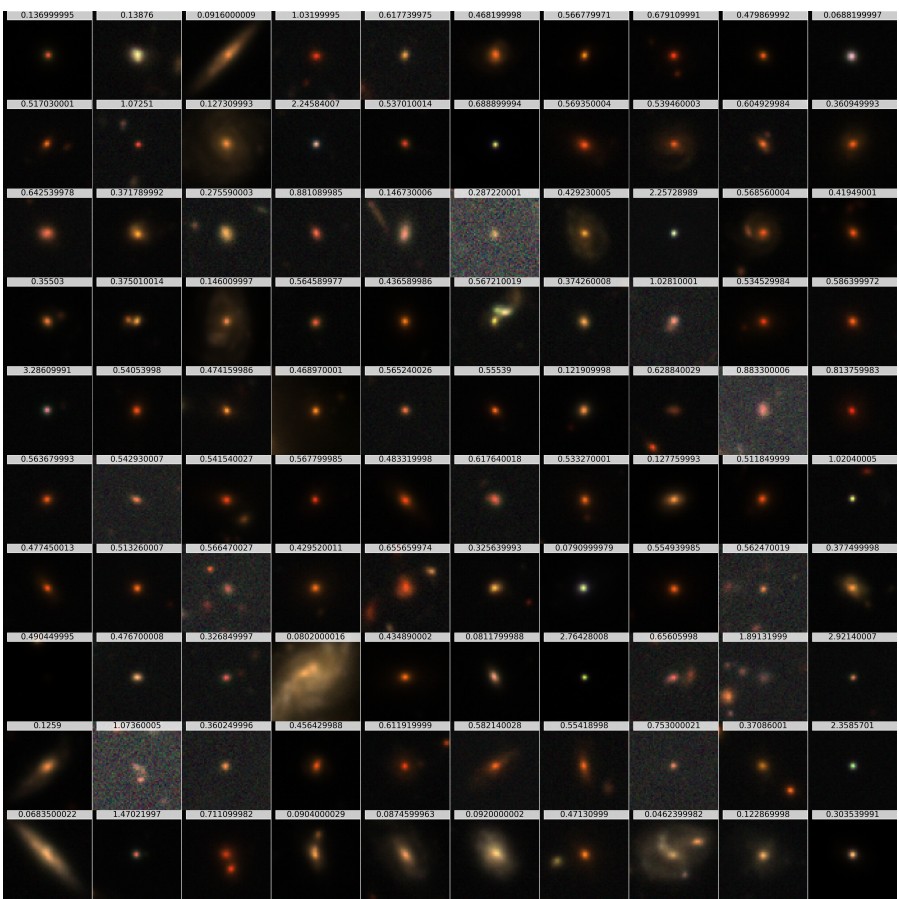

Figure 10: DDPM (False Color) Galaxies Generated at non-cherry-picked redshift values.

## C.2 DYNAMIC GALAXY GENERATIONS

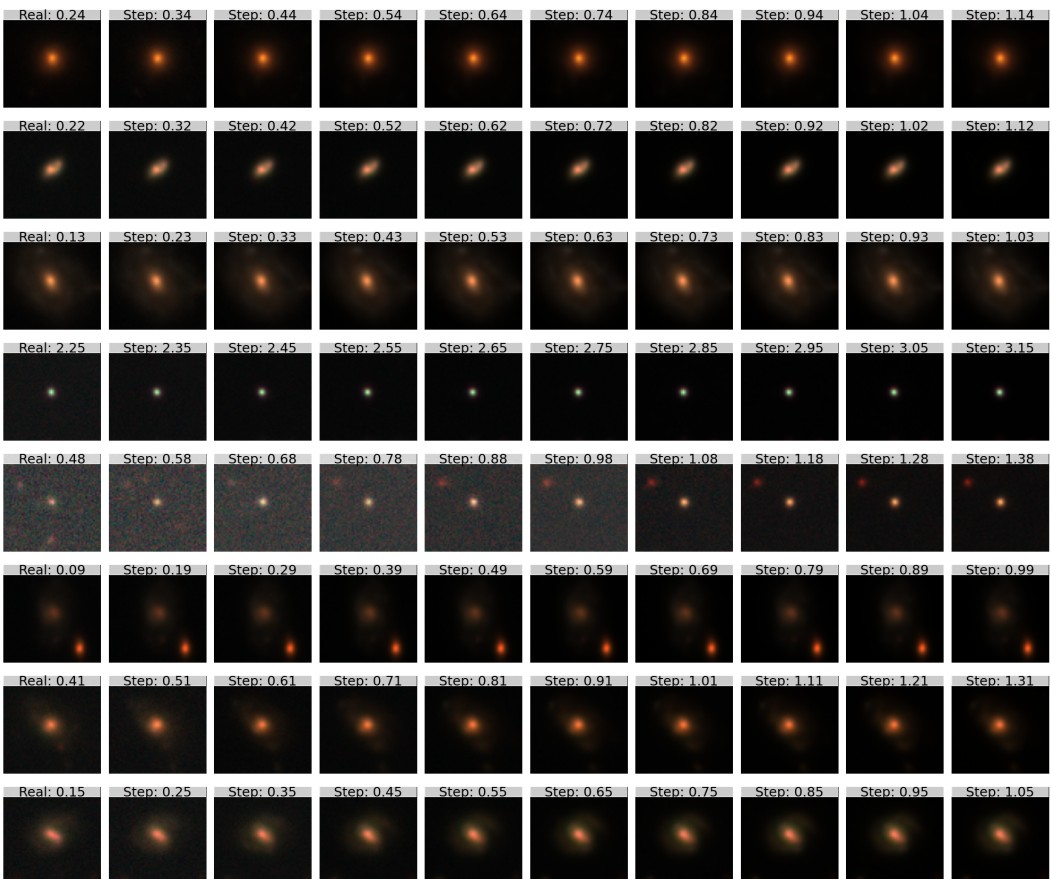

Figure 11: Real images and their corresponding trajectories from Algorithm 1

