# OpenReview forum: "Conditional Trajectories in Diffusion Models - Modeling Galaxy Evolution from Redshift"
_ICLR.cc/2025/Conference — ICLR 2025 Conference Withdrawn Submission_

### Official Review · Reviewer_VGou · 2024-10-22

**Soundness:** 2
**Presentation:** 1
**Contribution:** 2
**Rating:** 3
**Confidence:** 4

**Summary:**

This work proposes a strategy to simulate evolution of galaxies by using only galaxy images and redshift data. First, a conditional DDPM is trained by conditioning on the redshift values (in addition to timesteps). This paper makes certain assumptions to enable generation of galaxy images at different redshift values: KL divergence of distributions that differ by a small value of redshift is 0, bounded assumption of the gradient of reverse process w.r.t. the redshift value $z$, and that the diffusion model learns the reverse process correctly. The actual process of trajectory reconstruction involves running both forward and backward  diffusion process for each step of desired trajectory. The validity of this algorithm is justified by showing that the cumulative error is bounded based on a first-order Taylor approximation is used to predict noise at a slightly different redshift value $z + \Delta z$.

**Strengths:**

- The idea of simulating evolution of galaxy with diffusion model is novel.
- The proposed methodology to generate trajectories is easy to understand and has been explained well.

**Weaknesses:**

- Writing: The writing of this paper needs improvement.
    - The paper should include a paragraph that gives a brief overview of DDPM including the relevant notation. Ideally a paper should be self-contained. Only readers who are familiar with the notation in DDPM can follow this paper currently. Define the symbols for Beta Start and Beta End (Line 155)  in the overview of DDPM.
    - Line 107:  Define the filters, and perhaps include some example galaxy images at different redshift values + these filters as well in the appendix, so that the readers get an idea of what the training data looks like.
    - Define the metrics used to measure quality of generated galaxy images and mention why it is relevant in the Appendix (as opposed to mentioning these metrics in passing in Line 177). Also, indicate if these are ordered metrics - i.e. are higher values better or vice versa for some these.  Further, indicate reliability of these metrics and discuss consequences of high variance seen for these metrics in generated trajectory (Line 414, Fig 9).
- The primary contribution this paper is the method to generate trajectory of galaxy evolution at different redshift values. It is centered on the assumption that $KL(p(X|z) || p(X|z + \Delta z)) \rightarrow 0$ as $\Delta z \rightarrow 0$. However, in practice, $\Delta z$ is not small. For instance, a redshift value of $z=1$ is not close to 0. In fact, the step size will increase as trajectory increases. Therefore, I’m unsure about the validity of assumption, especially because the paper also notes that the accumulated error of the generated images increases along the trajectories. Further, the idea of conditional DDPM, where conditioning is on redshift values is not completely novel too, as this was previously explored in Li et al. 2024.
- I would argue that the proof in A.1.2. needs more rigor. It currently bounds the error in moving from $X_{t-1}$ to $X_t$. We also need to bound how the error accumulates along the trajectory (with a uniform bound). Further, given that this is continuous trajectory, the evolution of $X_t$ according to diffusion SDE needs to be accounted in the error calculation. The proposed algorithm runs a complete forward diffusion and backward diffusion at each z, therefore additional error will be accumulated due to the dynamics of diffusion SDE and it needs to be accounted.

**Questions:**

1. What is the value of $N$ (the number of trajectory timesteps) and $\Delta z$ used in the experiments? Did the authors try different values of $N$ and $\Delta z$ to understand how these values affect the quality of trajectories?
2. Figure 6 is difficult to interpret. Could the authors add more details of these experiments? For instance, how many trajectories were considered? Further, provide images of successful and failed trajectories in appendix. Similarly, in Figure C.2, can the authors indicate redshift errors for the images?

---

### Official Review · Reviewer_UneD · 2024-11-03

**Soundness:** 2
**Presentation:** 1
**Contribution:** 1
**Rating:** 3
**Confidence:** 4

**Summary:**

The authors focus on the problem of conditional generation in diffusion models when the conditioning variable is continuous, applied to astrophysics data, such as galaxy images with redshift values  (representing galaxy ages - a continuous variable) as the conditioning variable. They named their approach Conditional Trajectories on Diffusion Probabilistic Models (CTDM) which allows modeling galaxy evolution. A particular objective is the achieved smoothness in continuous conditional generation. This is accomplished by smoothing the conditioning signal through the addition of Gaussian noise and the application of positional encoding, which allows the construction of trajectories as the redshift values are varied smoothly.

**Strengths:**

*Interesting problem* - Achieving fine-grained control over a continuous variable in conditional generation is an intriguing problem. The authors attempt to address this by smoothing the conditioning signal through the addition of Gaussian noise.

**Weaknesses:**

- *Insufficient Citations and Background Discussion* :  The paper’s background and related work sections could be strengthened by including foundational references and prior works. For instance, the original paper on diffusion models [1] is missing.   [2] addresses similar challenges of continuous conditional generation, (please add in line 43).

- *Lack of Novelty in Conditioning and Training Techniques*: The contributions, while valuable, are somewhat limited, as they primarily reiterate the standard goals of diffusion models in modeling data distributions, whether conditional or unconditional. The use of Exponential Moving Average (EMA) during training, while effective, is a widely adopted optimization technique that does not add significant novelty to the approach. To enhance originality, the authors could explore alternatives to improve the conditioning signal.

[1] Dickstein et al., Deep unsupervised learning using nonequilibrium thermodynamics

[2] Ding et.a., CCDM: Continuous Conditional Diffusion Models for Image Generation. 2024

**Questions:**

1. Could you clarify in the text from the beginning what do you mean by "conditional trajectory-based generation"? This term could be misleading, as the reverse diffusion process itself constructs a trajectory, and conditioning on information naturally leads to a conditional trajectory.

2. Could you elaborate on what is meant by "stability of generative models under small perturbations" in the context of diffusion models? This concept differs from stability in VAEs, where learning robust and meaningful latent representations is often desired. In diffusion models, we don't really learn a latent space. The latent space that you mentioned is given by the way the conditioning signal is encoded right? rather than in the diffusion model.

3. How does your method compares against other approaches, for example against the work from [3]. It would be illustrative to just have a simple benchmark for example without adding the gaussian noise to $z$ and compare the performance of the model, also using other embedding techniques such as Fourier Encoding.

4. Why was sinusoidal encoding chosen over Fourier encoding? Sinusoidal encoding, initially introduced for NLP, is typically applied to discrete signals, whereas Fourier encoding provides higher expressiveness for continuous signals. Could Fourier encoding potentially provide smoother transitions in the conditioning variable, especially since your model aims for fine-grained control?

5. When constructing the trajectory, is $\Delta z$ added at each forward step/update in the forward process? If so, after $T$ steps, the image would reach a state of pure noise, simply due to the Brownian increments, so at that point the solution to the SDE would transition to another state which does not have any relation to the next step in the trajectory. Could you elaborate more on this, or am I missing something? Thanks!

[3] YQ Li et. al. Using Galaxy Evolution as Source of Physics-Based
Ground Truth for Generative Models

---

### Official Review · Reviewer_EStH · 2024-11-04

**Soundness:** 2
**Presentation:** 2
**Contribution:** 2
**Rating:** 3
**Confidence:** 4

**Summary:**

This paper studies the use of diffusion model to generate galaxies accross different values of redshift. Authors use redshift as a continuous conditionnal variable for their diffusion models. They study how well the diffusion model respects the conditionning and other galaxy related statistics. They then introduce a new method to generate trajectories of galaxies across different redshifts.

**Strengths:**

* According to the authors, it is the first time that diffusion models with continuous redshift conditionning have been studied
* The idea of studying a galaxy trajectory accross different redshift values is interesting, especially to explore regions of the galaxy space where data is missing
* They study how well conditional diffusion models preserve both their conditionning (in this cade redshift) but also other metrics important for galaxies: ellipticity, sersic index...

**Weaknesses:**

* Algorithm 1 is not consistent with the results. From my understanding of the provided code (perturb_redshift_iterative function), it seems that authors are starting with $X_T^{s+(n-1)\Delta z}=X^{s+(n-1)\Delta z}$. This initialization if out of distribution for the backward process. If this is indeed what the authors are doing, it would explain structural similarities between images from the same trajectory (but it does not make sense from an ML/generative modelling point of view).
* In scientific applications, providing estimates for sources of errors is crucial. Appart from the theory part, authors do not measure (or calibrate) the impact of the Huber loss in the denoising score matching objective (which is bound to induce some bias, especially at large noise level/large timesteps).
* No comparison with existing baselines (even with other generative model architectures like GANs that authors mentionned, or with discretized redshift and diffusion models)
* Conditionning on a continuous variable lacks novelty. Authors mention that "few studies have explored the behaviour of thses models when the conditionning variable is continuous". They do not provide any reference. For example in audio we have Audio Generation with Multiple Conditional Diffusion Model, Guo et al. AAAI 2024. For text to images generation: text tokens are usually encoded into a conditionnal vector (albeit the conditionning is different here). In scientific applications, diffusion models are often conditionned on physical paremeters (like redshift in this case).

**Questions:**

1. In my opinion, the way Algorithm 1 is described is not consistent with the sampled images of figure 8 or 9. In the forward diffusion step, when going from $X^{s+(n-1)\Delta z}$ to $X_T^{s+(n-1)\Delta z}$, you erase (or are supposed to erase) all information from $X^{s+(n-1)\Delta z}$, because any $X_T$ is supposed to be pure Gaussian noise (with zero mean). The fact that all your sampled trajectories are "geometrically" consistent (galaxies features at the same position in successive images) is not possible with this forgetting of information. My questions are then as follow:
a. Are you using the same Gaussian samples for each trajectory (same $X_T$ and then same gaussian increments used in the backward time process)?
b. If this is the case then aren't you "just" sampling in parrallel with the same seed different conditional trajectories of your diffusion model?
c. In the Forward Diffusion part of Algorithm 1, why not do that in one step? The forward process can be straightforwardly sampled from any timestep $t$ to any timestep $s$ ($s\geq t$) in one step.
d. Looking at the code (see the weakness section), it seems that the actual processed sampled is very different from both what I am describing here and what is written in algorithm 1. What is the correct algorithm?

2. For a scientific application, control of error bounds is important. The Appendix brings some element to that (appart from the missing information/misunderstanding as exposed in 1.). If my understanding of the algorithm that you are using is correct, what is the impact of the different steps on calibration of the results (both in term of generated images and downstream metrics)?

---

### Official Review · Reviewer_XS18 · 2024-11-04

**Soundness:** 3
**Presentation:** 2
**Contribution:** 2
**Rating:** 5
**Confidence:** 3

**Summary:**

This paper explored a new application of conditional diffusion model on galaxy data, which conditionally generate galaxy images based on red shift values. Through the marginal distribution of some physical index of the images, they validated that the diffusion model indeed learned the marginal and conditional distribution of the (redshift, galaxy images) . Interestingly, they devised a sampling algorithm can synthesize smooth “evolution” of galaxies based on changing conditional variable of redshift, and synthesized some galaxy evolution sequences which is impossible to observe directly.

**Strengths:**

- Conditional diffusion model based on continuous input values is indeed under-explored, and using them in a scientific setting (e.g. astronomy) is innovative and laudable. I feel more research using this type of generative models could be done in multiple domains.
- Fig 3, 4 showed pretty convincingly the diffusion model can capture the “natural statistics” of galaxy images.
- The sampling algorithm of the smooth trajectory indeed produced a smooth trajectory of galaxy evolution across redshifts, which no one can observe physically.

**Weaknesses:**

- “*Continuous trajectory **reconstruction***” is a little bit misnomer.
From what I understand of the proof in A.1.1, A.1.2, I can see that you can **construct** a smooth trajectory corresponding to changing $z$ values from an initial image, but I don’t think it’s **reconstruction** of any particular trajectory. The proof doesn’t show the construction have to correspond to any physical trajectory other than it’s smooth. “Reconstruction” sounds like it’s reconstructing a physical trajectory of evolution, which could be misleading to uncareful readers. The authors should consider change "reconstruction" to "construction" throughout, unless they can provide additional justification for using "reconstruction".
    - If the authors do want to claim “reconstruction”, another proof may be necessary: like given what assumption about the distribution of trajectories occurring physically, your construction will correspond to the “true” one.
    If not, the authors may want to tune down the phrasing slightly.
- Current sampling algorithm for continuous trajectory is smooth in deed, but very much heuristic. But **what exactly is it sampling from**? More concretely can you say sth. about what distribution is $X^{z+\Delta z}$ sampling from using your sampling algorithm? A more precise characterization will make the paper much stronger.
- In figure 6, 7, right panel, could you explain what exactly is being plotted on the y-axis? specifically how the tensor $\mu$ is converted to a scalar value, and why are negative values possible if it's the L2 norm of the gradient.
- Minor: Figure 6 is not very legible, authors can consider change the alpha value of the lines.
- The conditional mechanism seems a bit *ad hoc* and not thoroughly discussed. (See questions)
- A.1.1, A.1.2 the proof could be better structured with the proposition / theorem more clearly stated at front as a result.

**Questions:**

- **Conditioning mechanism**
Why adding redshifts $z$ and their embedding to the positional / noise / time encoding?
    - What’s the physical or machine learning reason to combine the red shift with the noise / time encoding $t$? Seems like there are many other ways without confounding these two inputs in the diffusion models.
    - Normally for categorical variables they will use a conditioning mechanism separated from noise / time.
- Being innovative indeed, could the authors discuss what types of physical insights / knowledge could be extracted from the constructed galaxy evolution trajectory? Currently the results seems more like self validation / sanity check, i.e. if we change the redshift conditioning signal, the redshift indeed changes.
- In figure 2, the scatter plot of ground truth and DDPM’s redshift have some grid / checker board patterns, what’s the source of that? is it a bias from the CNN model?

---

### Note · Authors · 2024-11-13

I have read and agree with the venue's withdrawal policy on behalf of myself and my co-authors.